# *Tersicoccus phoenicis* (Actinobacteria), a spacecraft clean room isolate, exhibits dormancy

Madhan Tirumalai,[1] Sahar Ali,[1] George E. Fox,[1] William Widger[1]

**ABSTRACT** Space missions or spacecraft equipment destined for sensitive environments, such as Mars, Europa, or Enceladus, are required to be designed to avoid forward contamination. Spacecraft are assembled in clean rooms (SACs) employing treatments to eliminate microbial contamination. However, some organisms can survive the cleaning procedures. Characterization of these populations, through both culture-based and sequencing methods, reveals that the majority consists of spore-forming bacteria. However, a smaller group of non-spore-forming organisms, primarily classified within the order *Micrococcales* of the phylum *Actinobacteria* (*Actinomycetota*), exists in some SACs. Despite their repeated occurrence and isolation, actinobacterial strains associated with SACs have not been studied for their dormancy potential. Here, we show for the first time that a non-spore-forming SAC isolate, *Tersicoccus phoenicis* (*Micrococcales*), enters dormancy under nutrient starvation. Dormancy in *Micrococcus luteus* involves a universal stress protein and a resuscitation-promoting factor (Rpf). Genes for these proteins are widely found in actinobacteria, including *T. phoenicis*. We show that dormant *T. phoenicis* (*Micrococcales*) can be revived through the addition of the Rpf to the media. Dormancy, as observed in the SAC actinobacterial isolate *T. phoenicis*, could well be a common trait adopted by other actinobacterial strains under the stressful conditions of spacecraft clean rooms or the ISS (International Space Station). This has implications for the persistence, identification, and recovery of such microbes from cleanroom facilities.

**IMPORTANCE** NASA's long-range goal of a human mission to the Mars surface raises issues relating to planetary protection. The concerns of forward contamination require that spacecraft assembly clean rooms (SACs) be maintained to inhibit microbial survival. However, despite these efforts, distinct microbial communities persist. Here, we show that a SAC isolate, *T. phoenicis*, exhibits dormancy, a state in which the cells are viable but not cultivable. Dormancy may help non-spore-forming organisms survive in the clean room environments utilized for space missions. This has implications for improving cleaning procedures.

**KEYWORDS** bacterial dormancy, resuscitation, spacecraft assembly facilities, clean rooms, human built environments, planetary protection, astrobiology

Planetary protection guidelines necessitate the implementation of rigorous cleaning and sterilization protocols in SAC facilities (1–4). Thus, the SAC environments are characterized by strictly controlled temperatures, oligotrophic conditions, filtered air circulation, humidity, sustained use of chemical disinfectants, and in some cases, ultraviolet radiation (UV) (5, 6). However, various microbial communities, predominantly endospore-forming *Bacillus* species, have been identified as survivors of these cleaning procedures (5, 7–22). Endospore-producing *Bacillus* sp. from SACs are posited to survive real space conditions (23). *Bacillus* strains and non-spore formers (such as *Deinococcus radiodurans*) are candidates for interplanetary transfer (24–26).

**Peer Reviewer** Gareth Trubl, Lawrence Livermore National Laboratory Physical and Life Sciences Directorate, Livermore, California, USA

Address correspondence to William Widger, widger@uh.edu, Madhan Tirumalai, mrtirum2@central.uh.edu, or George E. Fox, fox@uh.edu.

The authors declare no conflict of interest.

See the funding table on p. 12.

Non-spore-forming organisms, while less prevalent, have been identified in spacecraft assembly cleanrooms (SACs). However, our comprehension of their physiology and adaptability to the SAC environment remains limited (27–29). Some types of *Actinobacteria* (*Actinomycetota*), such as the *Micrococcales*, hold significance for cleanroom operations, planetary protection, and spaceflight (29–31). In the SACs for the OSIRIS-REx spacecraft and Mars 2020 missions, several new species of Actinobacteria (specifically *Micrococcales*) were identified, including six novel species (32, 33). Some isolates (*Arthrobacter koorensis*, *Paenarthrobacter nitroguajacolicus*, and *Mycetocola manganoxydans*) are capable of surviving drying, vacuum, and proton irradiation (34). *Micrococcus luteus* has been found in the ISS, on crew spacesuits, as well as in pharmaceutical clean rooms (27, 35–38).

Outside the SAC environments, Actinobacteria can undergo dormancy. This mechanism enables survival in the face of stressors such as nutrient starvation, extremes of temperature and desiccation, antimicrobials, or oxygen deprivation without the formation of spores (39, 40). Pathogenic non-spore-forming Gram-positive *Listeria monocytogenes* and the Gram-negative *Vibrio parahaemolyticus* are known to go into dormancy under stress and could remain undetectable in the dormant state (41, 42). The most studied example of bacterial dormancy is the pathogen *Mycobacterium tuberculosis*, which enters a state of clinical latency to evade the effects of antibiotics (43, 44). Dormancy refers to the Viable But Not Culturable (VBNC) state, in which cells are metabolically active but become uncultivable on agar plates, following nutrient starvation (45). VBNC bacteria generally remain non-cultivable under standard culture conditions and require the presence of resuscitation-promoting factors (Rpfs) (46), quorum-sensing molecules (47), or chemical agents (sodium pyruvate, catalase) (48). By contrast, persister cells are a dormant, phenotypically distinct subpopulation within a bacterial culture that tolerates antibiotics without acquiring genetic resistance (49, 50). Persisters can resume growth once antibiotics are removed (51, 52).

Cells in a dormant state can be induced to re-enter the growth phase through extended exposure to nutrient-rich media or by the action of an Rpf secreted by rapidly growing cells (46, 53, 54). Due to the challenges of studying latent tuberculosis, the non-pathogenic *M. luteus* has been used as a model organism (55). *M. luteus* has been isolated and revived from 125-million-year-old amber (56), indicating its ability to remain cultivable after extensive periods of stress.

In pharmaceutical clean rooms, *M. luteus* enters dormancy for prolonged survival (57, 58). However, the dormancy potential of SAC actinobacterial strains is poorly understood. NASA's standard spore assay (NSA) is a culture-based method targeting spores for quantification. Despite this, cleanrooms have low, persistent microbial loads (59). Only a small fraction of this microbial load is cultivable (60, 61). Uncultivable low microbial load is a challenge for planetary protection measures. Of particular interest are two very closely related actinobacterial organisms isolated from different SACs. *Tersicoccus phoenicis* strain 1P05MA$^T$ was isolated from the floor of an ISO 8 (3 520,000 particles > 0.5 µm m$^{-3}$, ISO 14644-1:2015) (62) SAC at the Kennedy Space Center (KSC), Florida, USA, while *T. phoenicis* strain KO_PS43 was recovered from the Centre Spatial Guyanais in the Final Assembly Building, Kourou, French Guiana (63, 64). This study focuses solely on *T. phoenicis* strain 1P05MA$^T$ from the KSC. We show that strain 1P05MA$^T$ enters dormancy under nutrient starvation. These results have implications for identifying and reviving actinobacterial strains from clean rooms. Furthermore, if dormancy turns out to be a major reason for their low detection, it could be a significant factor in planetary protection procedures involving the detection of *Actinobacteria*.

## RESULTS

### Evidence that *T. phoenicis* shows a dormant (VBNC) phenotype in AMM (acetate minimal media)

In *M. luteus*, VBNC is indicated (and thus measured) as a loss of colony forming units (CFUs) as the cells enter the stationary phase when grown in nutrient-limited media. As shown in Fig. 1A, the count of CFUs peaked at $5 \times 10^7$ per mL between days 1 and 2, subsequently declining by approximately one order of magnitude by day 4. By day 5, cell viability had declined to fewer than $10^2$ cells per mL, representing a decrease exceeding 5 orders of magnitude. By day 8, the decline had surpassed 7 orders of magnitude. Figure 1B illustrates that the OD (optical density) at 600 nm ($OD_{600}$) peaked at approximately 0.7 between days 1 and 2. This level was maintained until day 6, indicating that the cells are not dead despite showing fewer CFUs. Post-inoculation in AMM, the average $OD_{600}$ of cells for each experimental replicate ($n = 3$) was calculated at the specified days. The standard deviation is shown as error bars in Fig. 1. Comparison of the micrographs of exponentially growing *T. phoenicis* (Fig. S1(A)) and dormant *T. phoenicis* (Fig. S1(B)) shows no discernible difference in cell morphology.

Cleanroom environments are characterized by low humidity, constant air circulation, and nutrient scarcity, all of which can promote water loss and drying of microbial cells on surfaces. In fact, actinobacterial isolates from spacecraft clean rooms have been shown to survive drying/desiccation (34). We subjected suspensions of *T. phoenicis* to air-drying on sterile glass petri plates to simulate desiccation. In doing so, we aimed to determine whether *T. phoenicis* could survive such stress by entering a dormant (VBNC) state. Post air-drying, there was a marked $10^6$-fold reduction in cultivability (from $3.4 \times 10^6$ CFU to 1–4 CFU in 5 µL, a decrease of approximately 99.9%) within 48 hours post-drying (Table 1). *T. phoenicis* cells, when resuspended in plain water, demonstrated a loss of cultivability after 48 hours, even without being subjected to drying (data not shown). Doing similar studies with other actinobacterial isolates from such clean rooms would provide insight into how non-spore-forming bacteria may persist undetected in clean-room facilities despite extensive cleaning protocols.

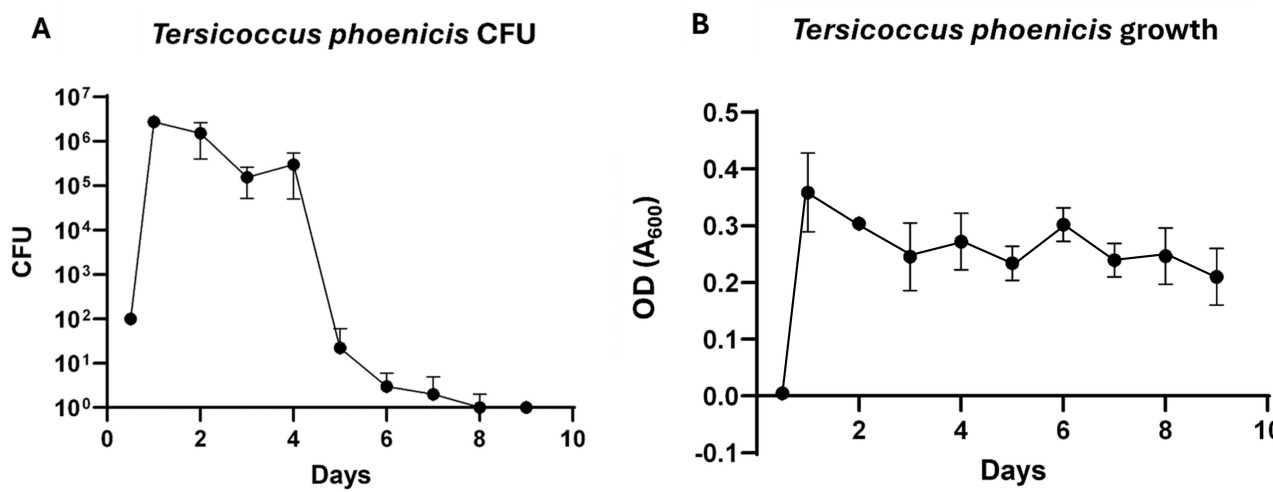

**FIG 1** (A) Viable colony-forming units (CFU) from growth in acetate minimal media (AMM). *Tersicoccus phoenicis* cells were pre-grown in LB media to approximately 1 $OD_{600}$ and diluted to 0.02 OD. A 1 mL aliquot was added to 50 mL in a 250 mL flask and grown at 30°C with vigorous shaking in triplicate. 1 mL aliquots were taken at the days indicated and serially diluted, and three sets of 10 µL of each dilution were plated on LB plates ($n = 3$). Spots that were countable were used to determine the CFU. Each set of technical replicates ($n = 3$) was averaged for each experimental replicate. The standard deviation is shown as error bars. As seen, the viable counts reached a maximum between 1 and 2 days a rapidly decreased after day 4. (B) The $OD_{600}$ of the cells for each experimental replicate ($n = 3$) was averaged at the indicated days after inoculation in AMM. The standard deviation is shown as error bars.

**TABLE 1** No. of CFUs (cell count)

| | Cell count (CFU/mL) | | |
|---|---|---|---|
| | Initial cell count | 48 hours (2 days) of drying | 168 hours (7 days) of drying |
| *M. luteus* | $3.15 \times 10^6$ | $7.52 \times 10^4$ | 2 (in 5 µL) |
| *T. phoenicis* | $3.4 \times 10^6$ | 1–4 (in 5 µL) | 1–4 (in 5 µL) |

## Estimation of Rpf concentration in the crude lysate

The total protein concentration of the crude lysate was measured to be 5.5 mg/mL using a Nanodrop spectrophotometer. SDS-PAGE analysis of the lysate revealed a prominent band corresponding to the predicted size of the full-length His-tag Rpf (~23–26 kDa) from *M. luteus* (65), and the expected expressed construct. Densitometric analysis estimated that this band represented ~50% of the total protein in the lysate. Based on this upper estimate, the concentration of Rpf was estimated at ~2.75 mg/mL (corresponding to ~1 µM Rpf in the lysate). If the proportion of Rpf was less, the effective concentration would also be reduced, suggesting that the observed activity may be underestimated. This estimation was used to standardize the addition of Rpf (lysate) to the growth media in both undiluted and serial dilution experiments. The presence of a single dominant band in the lysate, combined with the consistent activity of the lysate in resuscitation assays, supports the specificity of the expressed Rpf in the lysate. The SDS-PAGE gel used for this estimation is included as Fig. S2.

## Effects of Rpf on the growth of *T. phoenicis* and *M. luteus*

Dormant cells of *T. phoenicis* and *M. luteus* were grown in the presence and absence of the crude lysate from the *E. coli* overexpressing the Rpf from *M. luteus*. The layout of the 24-well microtiter plate is shown in Fig. S3. For this preliminary dormancy revival screening shown in Fig. 2, we tested each condition in a single well (i.e., $n = 1$), without any replicate(s).

The growth in Rows A and C (Fig. 2) containing LB (Luria broth) was more robust, reaching an $OD_{600}$ of ~0.8-1.0 compared to Rows B and D, containing acetate minimal media (AMM), which reached an $OD_{600}$ of ~0.4–0.5. In each row, the well (column 1) containing the undiluted Rpf (lysate) grew the fastest, that is, had the shortest growth lag phase of approximately 725 minutes (12 hours) in LB, and 1,350 minutes (22 hours) in AMM, for both the strains.

The lag phase decrease was dependent on the concentration of Rpf (lysate) present. Cell growth, in LB, in the absence of Rpf (lysate), (column 6) showed the greatest lag phase (Row A, approximately 2,300 minutes [38 hours] for *M. luteus* and Row C, 3,500 minutes [58 hours]) for *T. phoenicis*.

In AMM, *M. luteus* exhibited an extended lag phase lasting 3,500 minutes (58 hours) in the absence of Rpf (Fig. 2, Row B, column 6). By contrast, *T. phoenicis* in the absence of Rpf showed negligible growth in AMM without Rpf (lysate), defined as an $OD_{600}$ increase of <0.1 even after 5,000 minutes (83 hours), compared to an $OD_{600}$ increase of 0.35–0.4 in its presence. We also observe that the presence of diluted levels of Rpf (concentrations (1/1,000 dilution, ~1 nM, Fig. 2: columns 4–5) resulted in higher $OD_{600}$ values (~1.6–1.75) than seen in wells that have higher concentrations of the Rpf (lysate) (1.1–1.26). The growth rate of dormant cells of both strains was proportional to the Rpf (lysate) levels in the growth media (Fig. S4; Fig. 2). The growth rate could not be calculated when there was no discernible growth (in the absence of Rpf or low concentrations of Rpf) (Fig. 2), or when there was "spikiness" (due to cell aggregation).

In the absence of the Rpf (lysate), the paired t-test for the microbial growth indicated that there was a decrease in the growth, which was not statistically significant ($P = 0.08$). By contrast, in the presence of the undiluted Rpf (lysate), the test condition showed a statistically significant ($P = 0.01$, paired t-test) marked increase in microbial growth. Furthermore, an unpaired t-test revealed a highly significant difference ($P = 0.0003$), thus

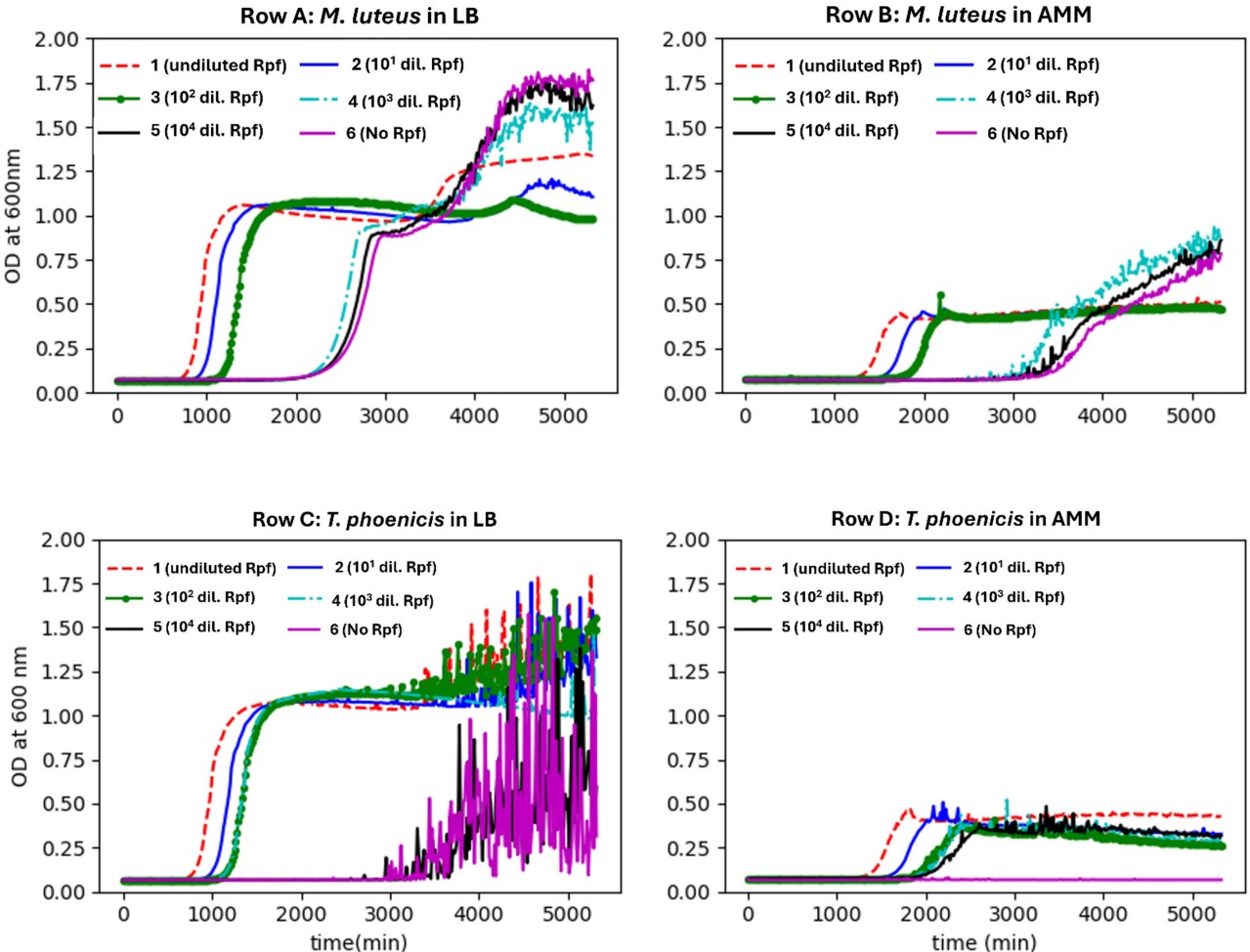

**FIG 2** Preliminary dormancy revival screening of the effects of Rpf (lysate of *E. coli* overexpressing the Rpf gene from *M. luteus*) on the growth curves of dormant cells of *Micrococcus luteus* and *Tersicoccus phoenicis*. This was a single study (*n* = 1) without replicates. Column 1 contained 10 µL of undiluted Rpf (~1 µM [27.5 µg/mL]), while columns 2–5 were serially diluted Rpf and column 6 (control) had no Rpf. AMM refers to acetate minimal media. Dormant cells of both strains showed spikiness after reaching a certain phase of growth. Dormant cells of *T. phoenicis,* in particular, showed increased spikiness in LB and negligible growth in the absence of Rpf in AMM. Growth was conducted and measured every 15 minutes in a 24-well (4 × 6) microtiter plate with 1 mL of medium per well using a Tecan SpectraFluor Plus instrument. Continuous shaking was maintained between successive readings.

confirming that the undiluted Rpf (lysate) promoted microbial growth relative to the control (absence of the Rpf (lysate)).

Figure S5A shows the relative lag time plotted against the serial dilution of Rpf (lysate). In Fig. S5B, the log of the dilution was plotted vs relative lag time and showed more detail. Rpf was effective at 100-fold dilution for *M. luteus*. By contrast, *T. phoenicis* appeared more sensitive and was effective at 1,000-fold dilution and did not grow in AMM in the absence of Rpf. In addition, the growth curves at later times of *T. phoenicis* in rich media show a strong "spikiness." An increase in cell clumping was observed as the cells enter dormancy (Fig. 4).

The growth study of *T. phoenicis* was repeated in triplicate in LB and AMM in the presence and absence of Rpf (lysate). In both cases, the first row contained an inoculum of growing cells of *T. phoenicis,* while the remaining three rows (*n* = 3) contained dormant cells of *T. phoenicis*. Column 1 did not have the Rpf (lysate), while column 2 had the undiluted lysate, and columns 3–6 had dilutions of 1/10, 1/10², 1/10³, and 1/10⁴. In LB, Rpf

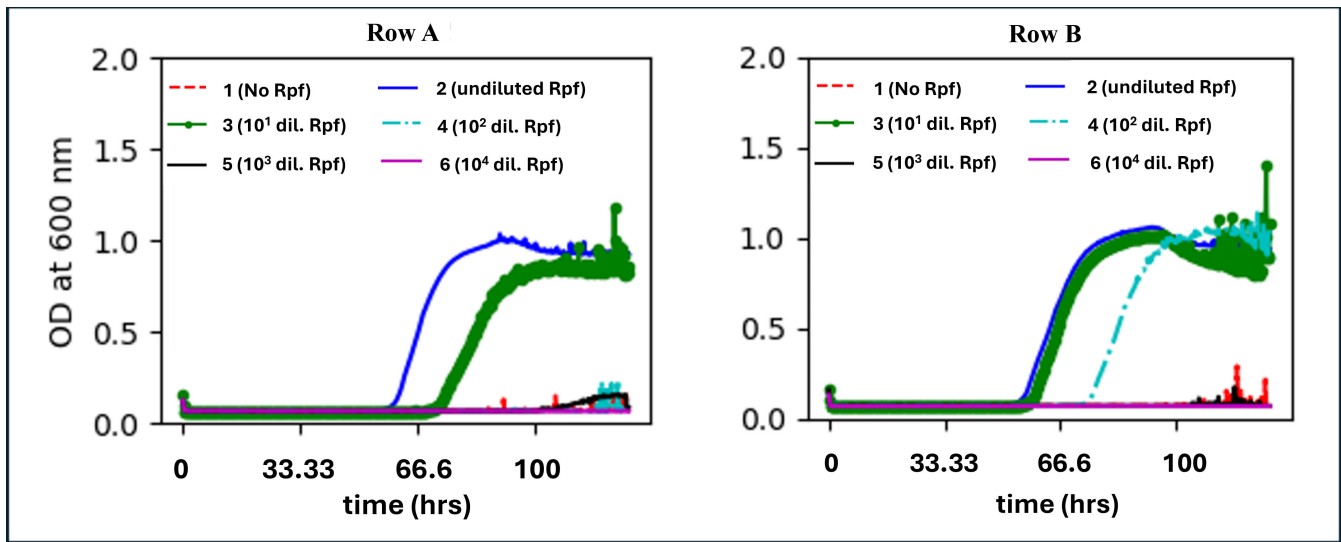

**FIG 3** The effects of Rpf (lysate of *E. coli* overexpressing the Rpf gene from *M. luteus*) on the growth curves of *Tersicoccus phoenicis* in acetate minimal medium (AMM). Rows A and B contained dormant cells of *T. phoenicis* (obtained after growth in AMM) (growth in duplicates). In both rows, Column 1 contained no Rpf (lysate), column 2 contained 10 μL of undiluted Rpf (lysate) (~1 μM; 27.5 μg/mL), and columns 3–6 contained serial 10-fold dilutions of Rpf (lysate) (1/10 to 1/10⁵). Growth was conducted and measured every 15 minutes in a 24-well (4 × 6) microtiter plate with 1 mL of medium per well using a Tecan SpectraFluor Plus instrument. Continuous shaking was maintained between successive readings.

(lysate) did not have any significant effect at any of the concentration(s) of Rpf (lysate) that were used (Fig. S6).

In AMM, because the growth was slow, the growth was allowed to go on for over 7,500 min (125 h/5 days). The longer duration resulted in the wells going dry in two rows (one row containing the growing cells and containing the first set of dormant cells), and hence is not shown. In the remaining two rows (A and B) of dormant cells, the undiluted Rpf (lysate) and dilution of Rpf (1/10) enhanced growth, while the lag increased from ~3,700 minutes (62 h) with undiluted Rpf to ~4,200 minutes (70 h) with Rpf (1/100) (Fig. 3). Thus, Rpf (lysate) clearly showed a growth-enhancing effect. At further dilutions of the Rpf (lysate), there was minimal enhancement of growth or no growth at all.

## Effects of air-drying (desiccation) on the cultivability of *T. phoenicis* on LB agar plates

Cell suspensions of *T. phoenicis,* when air-dried for under 48 hours and then resuspended in LB agar plates, showed a 10⁶-fold reduction in cell count. By contrast, *M. luteus* showed a 10-fold decrease for the same duration (48 h) (Table 1). After 7 days of air-drying (168 h), *M. luteus* also showed a 10⁶-fold reduction in cell count.

## Effects of Rpf on the growth of *T. phoenicis* air-dried (desiccated) for 7 days (168 h)

Cells of *T. phoenicis* were air-dried for over 7 days and resuspended in AMM in the presence and absence of the undiluted Rpf. They recovered growth with a lag phase of ~1,850 minutes (31 h) and ~3,000 minutes (50 h), respectively (Fig. S8).

## DISCUSSION

Representatives of diverse genera, including *Acinetobacter*, *Bacillus*, *Corynebacterium*, *Escherichia*, *Flavobacterium,* and fungi, have been isolated from spacecraft cleanroom facilities (8, 14, 66, 67). Many of them have been reported to be resistant to desiccation, UV, nutrient limitation, cleaning agents, and metals (68, 69). Understanding the growth

dynamics of microorganisms found on spacecraft surfaces is critical for developing effective cleaning protocols (8, 14, 30). When adaptation to the SAC environment does occur, microbiomes become niche-specific. For example, increased confinement and cleaning are reported to directly correlate with a loss of microbial diversity, which transitions from Gram-positive bacteria (*Actinobacteria* and *Firmicutes*) to Gram-negative (*Proteobacteria*) (70). A strong correlation between the succession of microbial diversity and human habitation has been shown during a 30-day human occupation of a simulated inflatable lunar/Mars habitat (71). The results presented here are relevant to addressing the challenges of understanding and combating the unique microbial diversity found in SACs.

Dormancy is thought to be a very ancient process that evolved from the early Earth (72). In SAC clean rooms, dormancy is often linked/attributed to the endospores of the *Bacillus* species, which undergo sporulation as a survival mechanism (73–75). Despite their repeated occurrence and isolation, SAC-associated actinobacterial strains have not been studied for their dormancy potential. While the dormancy traits of wild-type *M. luteus* are well characterized, our results show that the SAC actinobacterial isolate *T. phoenicis* also exhibits dormancy under nutrient starvation. The ability of the SAC isolate *T. phoenicis* to go dormant could be a key factor contributing to its undetectability in clean room environments.

The loss in cultural colony growth observed for the SAC isolate *T. phoenicis* (when grown in AMM media, Fig. 1) is also seen in many actinobacteria, including *M. tuberculosis* (44) and *M. luteus* (55). The slow growth of both *T. phoenicis* and *M. luteus* observed in wells containing AMM (Rows B and D in Fig. 2) suggested the AMM was nutrient-limited. Combined with no morphology change (Fig. S1A and B), these results clearly show that *T. phoenicis* can exhibit a VBNC phenotype under nutrient limitation. These findings are consistent with a dormant-like state in Actinobacteria, defined by lack of replication but retention of viability until triggered by resuscitation-promoting factors (76, 77). While alternative explanations, such as persistence or cellular injury (78, 79), cannot be entirely ruled out, the resuscitation phenomenon is often considered evidence of VBNC existence (80). Thus, the ability of these cells to resume growth upon exposure to the Rpf (lysate) supports this interpretation.

Actinobacterial growth is dependent on the presence of the Rpf (54). This has been demonstrated in *M. luteus* (77, 81). The gene sequence for the Rpf gene in *M. luteus* (locus tag DQN81_RS00545/Mlut_14360) has a homolog in the genome of the SAC isolate *T. phoenicis* with the locus tag BKD30_RS13480. Since the gene is already present in *T. phoenicis*, we hypothesized that cell dormancy of *T. phoenicis* should be reversed by the addition of Rpf, which initiates cell growth.

The Rpf (lysate) gene from *M. luteus* was overexpressed in an *E. coli* strain, and the crude lysate from the same was used for revival experiments. Crude lysate containing Rpf (lysate) has been shown to effectively resuscitate VBNC *Rhodococcus* sp. DS471 (82). Another study explored the use of *M. luteus* supernatant containing Rpf (lysate) to increase the cultivability of soil bacteria on oligotrophic media (83).

*T. phoenicis* and *M. luteus* cells were grown in minimal media (AMM) until they became non-cultivable. The fastest growth revival of non-cultivable cells of *T. phoenicis* and *M. luteus*, in the presence of the highest concentration of the Rpf (Fig. S4 and S5), clearly demonstrated that the cells were dormant and not "dead." The resumption of growth in column 6 in the Rows A and C (Fig. 2), containing LB in the absence of Rpf, suggested that the lack of Rpf expression limits growth until a threshold concentration is reached, in which growth resumes. In addition, we attribute the "spikiness" and increase in the OD in the absence of Rpf to agglutination (clumping) of cells (Fig. 4).

When suspensions of *T. phoenicis* cells (obtained from exponentially growing cells in rich [LB] media) are air-dried for 48 hours (desiccation), only a fraction of the cells are cultivable (a $10^6$-fold decrease in cultivability) on LB agar plates (Table 1). This is indicative of rapid entry into dormancy as a response to nutrient depletion and desiccation. When cells of *T. phoenicis* were air-dried for 48 hours and grown in AMM

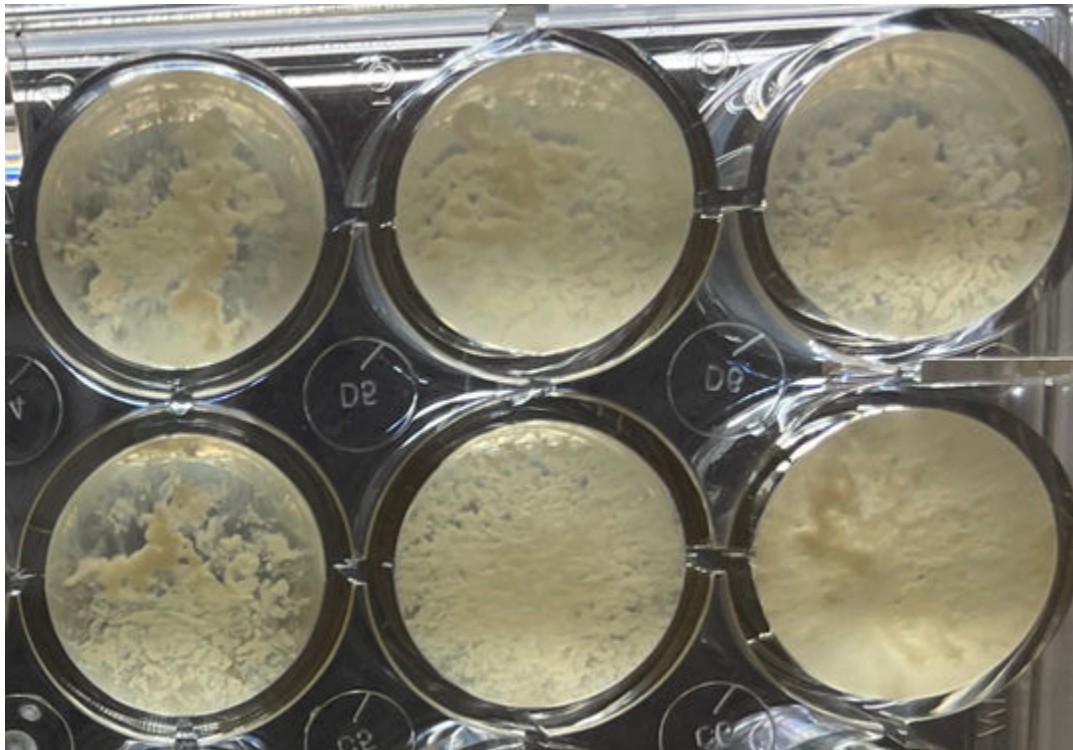

**FIG 4** *T. phoenicis* cells growing in "clumps" in acetate minimal media, shown after growth was terminated (~5,000 min) and plates were removed from the Tecan SpectraFluor Plus instrument.

without Rpf, they resumed growth with a short lag phase of ~1,020 minutes (17 hours) (Fig. S7). However, when they were air-dried for 168 h (7 days), the lag phase increased to ~3,480 minutes (58 h) (Fig. S8, Column 1). We speculate that while the process of air-drying the cells promotes dormancy within 48 h, it does not entirely inactivate the resuscitation-promoting factor. This could explain why the cells air-dried for 48 h could recover growth faster when placed in liquid media than those desiccated for 168 h. When cells air-dried for 168 hours were grown in AMM in the presence of undiluted Rpf containing crude lysate, the growth recovered with a shorter lag phase of ~1,850 min (31 h), compared to the 58 hours of lag phase in the absence of the Rpf (lysate). This clearly demonstrated that even after 168 hours of desiccation, the cells may be in a dormant state rather than dead (Fig. S8B). Overall, the lysate promoted modest growth compared to the negative control (absence of Rpf [lysate] containing crude lysate shown as a black line) (Fig. S8B).

Furthermore, we observed that when *T. phoenicis* cells that are actively growing in LB are resuspended in plain water, they show signs of entering dormancy while in the water suspension for more than 48 hours ($10^3$-fold reduction in the number of culturable cells), even without subjecting them to desiccation.

For *M. luteus*, there was only a 10-fold decrease in the number of cells that could be cultivable on LB agar plates (compared to the $10^6$-fold decrease for *T. phoenicis*) after 48 hours of air-drying. However, this decrease is $10^6$-fold after 168 h (7 days) of air-drying of *M. luteus* (Table 1). What could account for this difference in cultivability between *T. phoenicis* and *M. luteus* following desiccation? We posit that the difference is likely because *T. phoenicis* is a spacecraft isolate already pre-adapted to such conditions (nutrient-limited and desiccation). Hence, its response to the same is observed in its ability to readily go into dormancy.

On the other hand, *M. luteus* used in this study is a type strain that is not pre-adapted to desiccation and hence takes a longer time to go into dormancy. It remains to be seen

if other actinobacterial strains, including *M. luteus* isolated from cleanroom facilities as reported earlier (27), are capable of readily entering dormancy under similar conditions. Since the SAC isolate *T. phoenicis* enters dormancy under nutrient starvation, and it shares the universal stress protein(s) and the resuscitation-promoting factor found in *M. luteus* and *Mycobacterium tuberculosis*, the pathway toward dormancy is likely to be similar.

Although *M. luteus* is widely used as a model for dormancy studies due to its genetic and physiological parallels with *M. tuberculosis*, our findings suggest strain-specific variation in dormancy onset under desiccation. The type strain of *M. luteus* used here, not being a cleanroom isolate, exhibited delayed dormancy compared to *T. phoenicis*, which showed rapid loss of cultivability indicative of a dormancy response. This could reflect pre-adaptation of *T. phoenicis* to the nutrient-limited, low-humidity conditions of spacecraft cleanrooms. To confirm this, further studies with other cleanroom-derived actinobacterial strains are warranted, as supported by recent findings of desiccation- and radiation-resistant cleanroom isolates (29, 34).

Overall, in the more realistic scenario of the clean rooms or the ISS, cells from such strains may enter dormancy under nutrient starvation and desiccation (dry conditions). This can hinder their recoverability using conventional culturing methods. Methods like PMA (propidium monoazide) treatment can indicate the presence of DNA. However, azides only stain the DNA from dead cells that have been broken up (84, 85). Combining cultivation methods with propidium monoazide PMA-dye viability qPCR (v-qPCR) has helped detect VBNC-*Campylobacter* (86). Likewise, combining PMA and loop-mediated isothermal amplification (LAMP) has facilitated the detection and quantification of VBNC *E. coli* O157:H7 and *S. enterica* in fresh (food) produce (87). In the context of the HBEs of the SACs and the ISS, genomic sequences have been obtained from PMA-treated samples to measure intact microorganisms.

*Actinobacteria* are increasingly featured as uncommon extremophiles in SAC clean rooms, despite their low abundance. Repeated occurrence of microbes like *Tersicoccus phoenicis*, which are resilient to conditions in SACs, reveals gaps in sterilization protocols, necessitating enhanced methods to prevent forward contamination. Persistent microbes could degrade spacecraft materials or instruments, requiring countermeasures to ensure mission success. A recent paper by Schultz et al. (29) highlights the biofilm formation potential of spacecraft cleanroom microbes. Current detection protocols are biased towards the identification of the more abundant spores and may exclude the less metabolically active dormant component of NASA cleanrooms (33). The ability of a SAC actinobacterial isolate *Tersicoccus phoenicis* to enter dormancy suggests that this trait may be common among similar strains from SACs or the ISS.

We used a crude lysate from an *E. coli* overexpressing the Rpf protein from *M. luteus*. Hence, this precluded a precise dose-dependent interpretation of the Rpf activity. Nevertheless, enriching growth media with purified factors such as the Rpf might be a novel means to enable the recovery of *Actinobacteria* that would otherwise remain dormant and undetectable using conventional means from HBEs, including SACs and other ecosystems. Thus, the data presented here are directly relevant to planetary protection protocols. A more thorough understanding of such SAC-associated actino-bacterial isolates could inform updates to planetary protection guidelines (COSPAR), shaping mission planning and clean room protocols (4). Finally, it would be worthwhile to examine how dormancy affects the ability of Actinobacteria to survive stress factors, including biocidal agents. This will enable a better understanding of how they could survive and respond to the unfavorable environments in other planetary bodies/envi-ronments. Understanding extremophiles, surviving harsh clean room conditions, could guide the design of life-detection instruments and biosignature searches (88, 89). Furthermore, studying microbial survival strategies (e.g., dormancy, spore formation, and DNA repair) drives innovations in biotechnology, sterilization, and antimicrobial development, with applications in food and medical industries.

## MATERIALS AND METHODS

*T. phoenicis* DSM 30849 (1P05MA) B-59547 (NRRL) was obtained from the ARS Culture Collection, U.S. Department of Agriculture, and routinely maintained on Tryptic Soy Agar (TSA) or LB media. *Actinomycetota* (*Actinobacteria*) are strict aerobes that typically cannot grow by fermentation. An acetate minimal media (AMM), where acetate is the only carbon source, was used to sustain growth. AMM was prepared as described by Mali et al. (55). A glyoxylate shunt was used to ensure no fermentation could occur. Using AMM provides the distinct advantage of rapid depletion of acetate as the carbon source, leading to nutrient starvation stress and dormancy.

### Evaluation of dormancy in AMM

Individual *T. phoenicis* colonies from an agar LB plate were grown at 30°C with vigorous shaking in three separate conical flasks, each containing 50 mL of AMM media. At various times during the growth, a 1 mL aliquot was taken and serially diluted. 10 µL of each dilution was placed/spotted on an LB agar plate and allowed to grow. Colonies were counted from sufficiently diluted spots (10 µL) to determine CFUs. The $OD_{600}$ was also taken from both the first and the 1/10 dilution, and readings were used to measure growth rates.

### Evaluation of dormancy of *M. luteus* and the SAC isolate *T. phoenicis* after desiccation

As a step toward assessing the ability of the cells to tolerate desiccation, cells of *T. phoenicis* and *M. luteus* growing in LB were harvested from the logarithmic phase of growth, washed thrice with buffer (100 mM KCl and 50 mM Tris), and resuspended in plain sterile de-ionized water. The suspension of cells in water was plated out on LB agar plates to obtain an initial estimate of the number of CFUs. From these suspensions, 100 µL of cells was spotted in triplicate on a sterile petri plate (three spots each for *T. phoenicis* and *M. luteus*). The plates were allowed to dry at 30°C for 8 days, following which 100 µL of sterile water was added to the dried-up spot(s), and the dried-up cells were resuspended using a disposable cell scraper. They were then estimated for the number of CFUs on LB agar to estimate cell survival and cultivability after desiccation. Cells were then resuspended in LB and AMM media, and growth was observed.

### Isolation of *M. luteus* Rpf

*M. luteus* NCTC 2665 was obtained from the American Type Culture Collection (NCBI genome accession number NC_012803). The expressed product of the Rpf gene has three domains: the secretory signal, the C-terminal LysM, and the Rpf domains (76). When the Rpf domain-only protein was expressed, it was found as an insoluble inactive protein (data not shown). Expression of the LysM plus the Rpf domains resulted in an insoluble product in the inclusion body (90). Therefore, the full-length gene of *M. luteus* Rpf, including the secretion sequence, *lysM,* and the *Rpf* domain, was isolated and ligated in frame into the NcoI site of pNIC28. This created an N-terminal 6-His tag. The DNA was amplified using a forward primer containing the NcoI site and a reverse primer that was blunt-ended. The plasmid construct was initially transformed into *E. coli* cells. Transformed cells were selected, and the colony that showed the correct restriction digestion was sequenced. A stock amount of plasmid was isolated, and aliquots were used to transform the BL21 pLysS overexpressing strain of *E. coli* (90). The overexpressing strain was grown to 0.6 $OD_{600}$ in a 50 mL culture and induced with 100 mM IPTG for 4–6 hours. Following this, the overexpressing *E. coli* strain cells were collected by centrifugation, resuspended in 5 mL of 50 mM Tris-HCl, 0.1M NaCl, pH 8.0. They were lysed by sonication, and cell debris was removed first by centrifugation. The supernatant (lysate) was additionally passed through a 0.2 µm syringe filter. The filtered lysate containing the Rpf was either used as such or the protein components of the lysate were precipitated using ammonium sulfate (91). Total protein concentration was

quantified using a Nanodrop. SDS-PAGE (12%) gels were run in MOPS buffer at 90V, 2.5 h, to determine overexpression in induced cells, compared to cells that were not induced with IPTG. The gel was then stained with Coomassie blue stain and destained with 10% methanol solution. Figure S2 shows the SDS-PAGE electrophoretic profile of the lysate from uninduced and induced cells. The induced cells show a large protein band at the molecular weight of the His-tagged Rpf.

This undiluted lysate was used as the stock Rpf preparation, with serial dilutions (1/10–1/10,000) prepared freshly for growth assays. No additional purification was performed beyond filtration, as the biological activity of the lysate was reproducibly validated in resuscitation experiments first using *M. luteus* and then with *T. phoenicis*.

## Growth curves for *M. luteus* and *T. phoenicis* in the presence of Rpf

The loss of viability could be caused by cell death or cell dormancy. To evaluate whether the cells are dormant, growth curves were measured in the absence and presence of varying amounts of *M. luteus* Rpf. Growth was conducted and measured ($OD_{600}$) every 15 minutes in a 24-well ($4 \times 6$) microtiter plate with 1 mL of growth media in 2 mL wells, using a Tecan SpectraFluor Plus instrument. Continuous shaking was maintained between successive readings. The lag times in each well in the microtiter plate from Fig. S5 was calculated from the intersection of the growth curve before the increase in growth and the steepest slope of the cell growth for each well. Relative lag time was measured by subtracting the absolute lag time for the highest amount of Rpf (column 1) from each of the other absolute lag times for each of the columns in that row. The absolute lag time for the highest concentration of Rpf in Row A, column 1 was 725 min. Subtracting this value from the other absolute lag times gives a relative lag time, where the first column lag time is zero and the lag increases until no Rpf is added. Since the lag should be a negative value, the sign was changed to a negative value.

Initial starter cultures were grown in the absence of Rpf in rich media (LB) or AMM, depending on the experiment. Exponentially growing cells grown in 50 mL of LB were adjusted to 0.020 $OD_{600}$. Each well of a 24-well (four rows with six wells in each row) microtiter dish containing 1 mL of either LB (rows A and C) or AMM (rows B and D) was inoculated with 10 µL of the diluted cells. Row A and Row B were inoculated with *M. luteus* as a control for Rpf-stimulated growth, while Row C and Row D were inoculated with *T. phoenicis*. Cells in Row A and Row C were grown in LB media, while Row B and Row D were grown in AMM. In column 1 for each row, 10 µL of stock solution of Rpf was added. The following columns (2–5) contain serially diluted Rpf (1/10, 1/100, 1/1,000, and 1/10,000) while column six was the control without the addition of Rpf. This protocol was repeated for dormant cells of *T. phoenicis* in AMM and LB in the presence and absence of the Rpf in triplicate. Row A contained an inoculum of growing cells of *T. phoenicis* in either LB or AMM media. Rows B, C, and D contained dormant cells of *T. phoenicis*. In all the rows, column 1 had no Rpf, column 2 had 10 µL of undiluted Rpf, while columns 3–6 were serially diluted (1/10, $1/10^2$, $1/10^3$, and $1/10^4$ Rpf).

## Genome of *T. phoenicis* and *M. luteus*

The draft genomes of *T. phoenicis* (NCBI genome accession no: NZ_MRDE01000076, NCBI RefSeq assembly: GCF_001968835.1, GenBank assembly: GCA_001968835.1) and *M. luteus* (NCBI genome accession number NC_012803, NCBI RefSeq assembly: GCF_000023205.1, GenBank assembly: GCA_000023205.1) were obtained from the NCBI database. BLAST searches (92) were performed for the Rpf gene.

## Tomocube imaging

The Tomocube HT-X1 is a quantitative phase imaging microscope that uses holotomography to produce 3D images based on refractive index differences within the observed sample (93, 94). This facility was provided as part of the Tomocube instrument demonstration at the Biology & Biochemistry Imaging Core at the University of Houston. Cell

suspension (100–200 µL) was placed on a Tomodish for imaging by the Tomocube system (Tomocube Inc., Republic of Korea). By acquiring multiple images using a 450 nm LED to illuminate the sample at various angles with patterned illumination, 3D tomograms were calculated to determine the refractive index of each voxel within the imaged volume. The resulting images have a lateral resolution as low as 156 nm and a depth of up to 140 µm. The X-Y dimensions of each image are 165 µm × 165 µm.

## ACKNOWLEDGMENTS

This work was supported in part by the National Science Foundation Award NSF-MCB-EAGER 2227347 to M.T. and G.E.F, and in part by the University of Houston's Drug Discovery Institute (DDI) Seed Grant awarded to W.W., M.T., and G.E.F.

The authors gratefully acknowledge Dr. Parag Vaishampayan (Space Biosciences Division, NASA Ames Research Center) and Dr. Aaron Regberg (Planetary Protection, NASA Johnson Space Center), whose critical reviews and insightful comments during the peer review process were instrumental in shaping and strengthening this manuscript. They also thank Brian Templin from Tomocube for help with obtaining images with the Tomocube HT-X1 system.

M.T. and W.W. conceived the experimental work. M.T. maintained the cultures, performed the growth and dormancy studies on T. phoenicis, and obtained the Tomocube images. S.A. and W.W. performed the growth and dormancy studies on M. luteus and the cloning of the Rpf gene. M.T., W.W., and G.E.F. participated in discussing the results during various stages and writing the finished manuscript. M.T. and W.W. prepared the figures. All authors discussed the results in the manuscript.

## AUTHOR AFFILIATION

[1]Department of Biology and Biochemistry, University of Houston, Houston, Texas, USA

## AUTHOR ORCIDs

Madhan Tirumalai http://orcid.org/0000-0002-5999-333X
George E. Fox http://orcid.org/0000-0001-7767-8387
William Widger http://orcid.org/0009-0005-8387-8074

## FUNDING

| Funder | Grant(s) | Author(s) |
| --- | --- | --- |
| National Science Foundation | NSF-MCB-EAGER 2227347 | Madhan Tirumalai |
| | | George E. Fox |
| University of Houston | University of Houston's Drug Discovery Institute (DDI) Seed Grant | Madhan Tirumalai |
| | | George E. Fox |
| | | William Widger |

## AUTHOR CONTRIBUTIONS

Madhan Tirumalai, Conceptualization, Data curation, Formal analysis, Investigation, Methodology, Project administration, Resources, Software, Supervision, Validation, Visualization, Writing – original draft, Writing – review and editing | Sahar Ali, Data curation, Formal analysis, Investigation, Methodology, Validation | George E. Fox, Conceptualization, Data curation, Formal analysis, Funding acquisition, Investigation, Methodology, Project administration, Resources, Supervision, Validation, Writing – original draft, Writing – review and editing | William Widger, Conceptualization, Data curation, Formal analysis, Investigation, Methodology, Project administration, Resources, Software, Supervision, Validation, Visualization, Writing – original draft, Writing – review and editing

## ADDITIONAL FILES

The following material is available online.

### Supplemental Material

**Supplemental figures (Spectrum01692-25-s0001.docx).** Fig. S1 to S8.

### Open Peer Review

**PEER REVIEW HISTORY (review-history.pdf).** An accounting of the reviewer comments and feedback.

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
