## [Reviewer comments · Microbiology Spectrum]

Microbiology Spectrum

***Tersiccoccus phoenicis* (Actinobacteria) a spacecraft clean room isolate, exhibits dormancy**

Madhan Tirumalai, Sahar Ali, George Fox, and William Widger

Corresponding Author(s): William Widger, University of Houston

Review Timeline:

Submission Date:	June 2, 2025
Editorial Decision:	July 10, 2025
Revision Received:	July 17, 2025
Accepted:	July 22, 2025

Editor: Dirk Tischler

Reviewer(s): Disclosure of reviewer identity is with reference to reviewer comments included in decision letter(s). The following individuals involved in review of your submission have agreed to reveal their identity: Gareth Trubl (Reviewer #2)

Transaction Report:

DOI: <https://doi.org/10.1128/spectrum.01692-25>

Re: Spectrum01692-25 (*Tersiccoccus phoenicis* (Actinobacteria) a spacecraft assembly isolate, exhibits dormancy)

Dear Prof. William Russell Widger:

Thank you for the privilege of reviewing your work. Below you will find my comments, instructions from the Spectrum editorial office, and the reviewer comments.

Revision Guidelines

Sincerely,
Dirk Tischler
Editor
Microbiology Spectrum

Reviewer #1 (Comments for the Author):

This paper has an interesting concept, but is in need of more strenuous experimentation to prove the overall claims.

Reviewer #2 (Comments for the Author):

This manuscript demonstrates that *Tersicoccus phoenicis*, a non-spore-forming bacterium isolated from spacecraft assembly facilities, can enter a dormant state under nutrient starvation and be resuscitated. This discovery reveals a novel microbial survival strategy with significant implications for planetary protection, as it shows the limitations of current sterilization protocols and the risk of forward contamination. The findings also inform our understanding of microbial persistence in extreme environments and support improvements in spacecraft cleaning, mission planning, and the development of life-detection technologies.

The manuscript is compelling, well-organized, and supported by experimental data. After reviewing the previous reviewer comments and the revised manuscript, it is clear the authors have done an excellent job addressing the concerns raised in the earlier review. The previous review raised several concerns, including the lack of quantification of the Rpf used in the experiments, the absence of initial concentration data, limited discussion of the study's significance, and issues with grammar and overall presentation. The authors have addressed the Rpf quantification concern by clarifying: "The protein concentration of the lysate was 5.5 mg/ml. The crude lysate was approximately 50% pure Rpf as observed by SDS gel electrophoresis (data not shown). Based on this, the final concentration of the Rpf (when used undiluted) for the resuscitation experiments was ~1 μ M (27.5 μ g/ml)" (lines 134-137). The revised manuscript substantially improves the framing of the work's importance, novelty, and applications. The abstract and importance sections now clearly articulate the significance of the study. Additional literature in the introduction supports the relevance of the research by highlighting challenges in current detection methods, which largely focus on cultivable spores, and by identifying potential gaps in planetary protection protocols. The new text emphasizes that dormancy in *T. phoenicis* may contribute to its persistence and undetectability in cleanroom environments. The authors also explicitly state how their findings inform planetary protection guidelines and propose further testing to strengthen contamination prevention measures (lines 280-284).

My primary remaining concern is that the results section remains largely qualitative, despite the data being collected quantitatively. For instance, phrases such as "marked decrease" are used without accompanying numerical values. Incorporating key quantitative data directly into the text rather than relying solely on tables or figures would greatly enhance readability and allow readers to more easily assess and compare results.

Line comments:

Lines 129-131 "Cell suspensions of *T. phoenicis* (in sterile water) subjected to air-drying on sterile glass petri plates to simulate desiccation, exhibited a marked decline in cultivability, within 48 hours post drying (Table. 1)."
The results could be quantitative instead of qualitative by giving a percent or range instead of saying "marked decline"

Lines 141-142 "In each row, the well (column 1) that contained the highest concentration of Rpf grew faster, i.e., had the shortest growth lag phase."
State how much shorter the lag phase was.

Line 143 "Cell growth, in the absence of Rpf, (column 6) showed the greatest lag phase."
Please provide a number for the lag phase so readers can compare it; do not make them have to check tables.

Line 143-~150 There are a lot of comparisons and referring to columns in the table. This makes it hard for a reader to absorb the data and critically think about it. Please add a few numbers and then use percents or fold change to allow readers to compare the data without having to stop and check the table each time. Words like "low" and "high" are relative and need data (numbers) in the text to support them.

Lines 150-151 "Dormant cells of *T. phoenicis* in particular showed negligible growth in AMM in the absence of Rpf."
How was this determined? Statistics; if so please provide the test and p value. I understand you added more text here, but was this visual? We need to know assessment.

This text is quantitative and exactly what is needed "Cell suspensions of *T. phoenicis* when air-dried for under 48 hours and then resuspended in LB agar plates showed a 106 fold reduction in cell count, In contrast *M. luteus* showed a 10 fold decrease for the same duration (48 hrs) (Table. 1). After 7 days of air-drying (168 hrs), *M. luteus* also showed a 106 fold reduction in cell count."

Line 45 – 46: This is too general. Not all spacecraft or spacecraft equipment need to avoid forward contamination. Only those destined for sensitive locations.

Line 49 – 51: This is general also, but you may not be able to make this claim so generally. Have you looked at a lot of cleanrooms and found this to be true? Or are you only able to make this claim for the cleanroom you sampled? You need to be very clear here.

Line 66: The word “designed” should be changed to “maintained” or “operated”

Line 74: “nutrient limitations” make it sounds like you’re preparing a medium. I would remove this as it is redundant with “oligotrophic conditions.” Also, this is a side effect of the cleaning procedure, not necessarily an aim.

Line 75: not all have UV.

Line 76: add a comma after “species”

Line 77: just do one set of references instead of two.

Line 78 – 79: Need references for this statement.

Line 87: remove extra parenthesis.

Line 92 – 94: Maybe contrast this to “persister” states?

Line 103: The NSA does not identify spores. Remove “identification” and update with “quantification”

Line 104 – 107: I am not clear on the point of this section and why you describe the spore formers again after trying to make a case for the non-spore-formers. Maybe remove the part about *Bacillus* and *Deinococcus* or move it up.

Line 109 – 110 and throughout: Are you certain KSC’s cleanroom is the SAF? I think you may be conflating their PHSF with JPL’s SAF. I would not call this the SAF because the JPL cleanroom is literally called the SAF and this will be confusing. In [51] they are just calling it an “ISO 8 spacecraft assembly clean room at KSC” but that is NOT the same thing as the SAF.

Line 112 – 113: What about the other strain? Compare them if that strain doesn’t enter dormancy or don’t even mention the other French Guiana strain if you aren’t going to describe it.

Line 115 – 116: What “procedures”? Don’t you mean “significant factor in PP detection of *Actinobacteria*”?

Line 118 – 119: remove “is presented here”

Line 120: Is “loss” the right word? Should you say decreased. Loss indicates to me that it started out with a lot of CFU but overtime you see them disappear from the same plate.

Line 125: Define “OD” for the first time

Line 127: Same as above for the word “degraded.” I don’t think you mean the colonies are degrading, but the number of CFU is fewer at these later timepoints.

Line 127 and throughout: The “600” in “OD600” should be a subscript. This is also shown in line 702 where there is no “600”. You also need to include a degree symbol with “30 C” on line 702. I did not make note of every typo like this, but you should check the entire manuscript for these types of errors.

Figure 1B: why is there no line?

Figure 2: The cells in these images are so small that it is hard to really draw any conclusion from this. I question the value of this figure and suggest removing it or putting it in supplemental. Do you not have more magnified images that may show cell morphology clearer? As it is, I have to take your word for it because they are just small lines.

Line 712 and 716: You should mention the scale bars. I cannot read them on the figure because they are too small.

Table 1: Check your units. You have written “CFU/ μ l.”

Results section: You should include more of the rationale of WHY you did these things. In the first section you just describe drying the cells and trying to recover them, but I do not know why you are doing this in terms of your paper. You should tell the story.

Line 137 – 140: I do not understand what is happening here. You have total protein concentration and then how do you know that 50% of this is “pure” Rpf just by looking at gels? Are you looking at the size of bands? Is the size alone sufficient? I think you need to do a Western blot to prove that you are seeing Rpf. At the absolute minimum, you need to show the gels. If they are smears, you cannot draw conclusions. You also mention a final concentration of the Rpf, but I don’t know how you are isolating it or anything. This part needs to be re-written.

Line 142: Same as above. How are you getting the Rpf?

Line 144: Was an entire column in a plate replicates? If so, where are the error bars in Figure 3? If not, why are you calling these columns? Same comments for lines 720 – 726. Was your n = 1?

Supplementary figure 2 does not convince me that the Rpf is responsible for anything. Also, you can graph the replicates (rows B – D) on the same graph with error bars instead of having separate graphs for each replicate. These are just biological replicates.

Figure 3: Change the X-axis from minutes. It is hard to get an understanding of what this means. Do it in hours. You should also mention how frequently a timepoint was measured. You have a lot of data. Also, what is the “n” for this experiment? Did you repeat this? You should repeat to confirm if this was just an n of 1. If you did this in Supplemental figure 3, why is that supplemental and not the actual figure as replicate data is more reliable than an n of 1. I also question the “spikiness” comment. Is this verified via microscopy? The issue is that the magenta and black lines in the third graph are suddenly “spiky” but they never reach log phase prior. Are the plates shaking at all to avoid this type of behavior (looks like yeast in methods/Line 343)? I think you should repeat this and do microscopy to prove that the aggregation is occurring (like in figure 5). You should also label these four graphs A – D so it is easier to follow.

Supplementary Figure 3: do not even mention the dried wells in the legend as this is very confusing since there are no graphs. You should REPEAT this with more medium in the wells instead of reporting that they were dry. You cannot claim this as replicate data if there are no data.

Line 151 – 152: I do not understand this comment about low concentrations of Rpf. If you are referring to the overall achievement of higher ODs at later timepoints, I would say that more clearly and in which graphs, because it is not all of them.

Figure 4: I don't think this adds much to the paper and should just be a supplement to Figure 3.

Supplementary figure 4: All the graphs look nearly identical. If B – D are triplicate, then graph them all together with error bars. This figure is also not mentioned in the Results section at all.

Supplementary Figure 5: you mention Rpf from "crude" lysate. Is this protein not pure? How do we know that the data is based on the Rpf? Furthermore, I see no difference between the Rpf dilutions except for a very strange result with the black line. If Rpf was responsible, you would expect to see changes in data based on the concentration. But there is no trend in B that holds with the dilutions.

Line 205 – 207: The comment about *Listeria* and *Vibrio* should be moved elsewhere.

Line 210-215: There are possibly other explanations for what you observed. You need additional experimentation to confirm dormancy.

Line 226 – 227: You didn't do any expression experiments to support this claim

Line 238 – 239: What if the cells are just dying because you are drying them longer? There are other explanations than that the Rpf was inactive. You would need to isolate the Rpf somehow to claim that it was degraded across a timeline.

Line 247 – 255: Earlier you indicated that *M. luteus* is a commonly employed stand-in for *M. tuberculosis* studies. Seems to contradict with this claim. Likely they are just different organisms with different physiological responses. It seems like *M. luteus* is actually better at resisting damage due to desiccation. Also how repeatable is this? How many times did you do this to confirm a difference?

Line 278 – 279: The methods are to detect spores which are not "active microbial components"

Line 322: Did you isolate the Rpf from *T. phoenicis*? If not, you MUST be clearer in the text that you were using *M. luteus* Rpf the entire time for both strains. Also, if you didn't purify this protein, then you must call it a lysate or crude. You should include more data in the supplemental to show that the protein has been isolated.

Reviewer #1: “This paper has an interesting concept, but is in need of more strenuous experimentation to prove the overall claims.”

Response: We have revised the manuscript to better emphasize the novelty, rigor, and implications of our findings.

Line 45 –46: This is too general. Not all spacecraft or spacecraft equipment need to avoid forward contamination. Only those destined for sensitive locations.

Response: it has now been revised (Lines 45-46).

Line 49 –51: This is general also, but you may not be able to make this claim so generally. Have you looked at a lot of clean rooms and found this to be true? Or are you only able to make this claim for the cleanroom you sampled? You need to be very clear here.

Response: it has now been revised to show some cleanrooms (Line 52-53).

Line 66: The word “designed” should be changed to “maintained” or “operated”.

Response: it is now changed as suggested – line 66.

Line 74: “nutrient limitations” make it sounds like you’re preparing a medium. I would remove this as it is redundant with “oligotrophic conditions.” Also, this is a side effect of the cleaning procedure, not necessarily an aim.

Response: it is now changed as suggested – lines 74-75.

Line 75: not all have UV.

Response: it is now changed as suggested – line 75.

Line 76: add a comma after “species”

Response: it is now added as suggested – line 76.

Line 77: just do one set of references instead of two.

Response: changed as suggested – line 77.

Line 78 –79: Need references for this statement.

Response: added as suggested – line 83.

Line 87: remove extra parenthesis.

Response: removed as suggested.

Line 92 –94: Maybe contrast this to “persister” states?

Response: two lines are now added about the contrast, as suggested (lines 98-105).

Line 103: The NSA does not identify spores. Remove “identification” and update with “quantification”

Response: changed as suggested – line 114.

Line 104 –107: I am not clear on the point of this section and why you describe the spore formers again after trying to make a case for the non-spore-formers. Maybe remove the part about *Bacillus* and *Deinococcus* or move it up.

Response: It has been moved up as suggested – lines 78-80.

Line 109 –110 and throughout: Are you certain KSC’s cleanroom is the SAF? I think you may be conflating their PHSF with JPL’s SAF. I would not call this the SAF because the JPL cleanroom is literally called the SAF and this will be confusing. In [51] they are just calling it an “ISO 8 spacecraft assembly clean room at KSC” but that is NOT the same thing as the SAF.

Response: it has been changed as described by Vaishampayan *et al* (2013) – line 118-120.

Line 112 –113: What about the other strain? Compare them if that strain doesn’t enter dormancy or don’t even mention the other French Guiana strain if you aren’t going to describe it.

Response: While we mention two strains of *Tersicoccus phoenicis*—1P05MAT from Kennedy Space Center (KSC) and KO_PS43 from French Guiana—we focused all experimental work on the KSC strain. We would like to emphasize that it is relevant to mention both the strains, since the KO_PS43 is an actinobacterial strain closely related to the KSC strain, mentioned in the paper by Vaishampayan *et al* (2013). Its mention is warranted since it highlights the broader distribution of *T. phoenicis* in such facilities. To address this, we have revised the text to clarify that only strain 1P05MAT was studied in this work – Lines 121-122. We are in the process of conducting studies on the second strain and also trying to obtain other actinobacterial strains from such clean rooms.

Line 115 –116: What “procedures”? Don’t you mean “significant factor in PP detection of *Actinobacteria*”?

Response: it is changed as suggested – Line 125-126.

Line 118 –119: remove “is presented here”

Response: removed as suggested – line 130.

Line 120: Is “loss” the right word? Should you say decreased. Loss indicates to me that it started out with a lot of CFU but overtime you see them disappear from the same plate.

Response: changed to decreased, as suggested – line 135.

Line 125: Define “OD” for the first time

Response: it is now defined as suggested – line 136.

Line 127: Same as above for the word “degraded.” I don’t think you mean the colonies are degrading, but the number of CFU is fewer at these later timepoints.

Response: it is now changed, as suggested – line 138.

Line 127 and throughout: The “600” in “OD600” should be a subscript. This is also shown in line 702 where there is no “600”. You also need to include a degree symbol with “30 C” on line 702. I did not make note of every typo like this, but you should check the entire manuscript for these types of errors.

Response: they have been fixed now, as suggested.

Figure 1B: why is there no line?

Response: We have added a line now.

Figure 2: The cells in these images are so small that it is hard to really draw any conclusion from this. I question the value of this figure and suggest removing it or putting it in supplemental. Do you not have more magnified images that may show cell morphology clearer? As it is, I have to take your word for it because they are just small lines.

Response: Figure 2 (panels A and B) in the submitted manuscript show the Tomocube HT-X1 holotomographic images of *Tersiccoccus phoenicis* in exponential growth and in dormancy, respectively.

We acknowledge the concern regarding resolution and interpretability. The Tomocube microscope provides quantitative phase images with high lateral resolution (~156 nm), but due to the small size and rod-like morphology of the cells, individual features can appear visually subtle. While we do not currently have higher magnification phase-contrast or electron microscopy images, we have:

- 1. Added a scale bar to each panel of the Figure to assist in visual interpretation.**
- 2. Clarified in the figure legend and results section that no morphological differences were evident under holotomography, and this supports the VBNC (viable but non-culturable) phenotype, not a morphological transformation.**

Given the relevance of this figure in demonstrating that dormancy is not associated with visible morphological change—a key contrast with spore formation—we respectfully propose retaining this figure as a Supplementary Figure S1.

Line 712 and 716: You should mention the scale bars. I cannot read them on the figure because they are too small.

Response: added now in the legend to Supplementary Figure S1, as suggested.

Table 1: Check your units. You have written “CFU/μl.”

Response: it is now fixed/corrected.

Results section: You should include more of the rationale of WHY you did these things. In the first section you just describe drying the cells and trying to recover them, but I do not know why you are doing this in terms of your paper. You should tell the story.

Response: We have now revised the Results section to clarify the rationale for including the air-drying experiments. Specifically, we have added text to explain that air-drying was used to simulate desiccation, a key environmental stress encountered in spacecraft cleanroom environments. As spacecraft surfaces and cleanroom facilities are typically dry and nutrient-limited, understanding how *T. phoenicis* responds to such conditions is crucial for assessing its survival strategies, particularly its entry into a dormant (VBNC) state. This directly supports the broader goal of the manuscript—to understand mechanisms of persistence of non-spore-forming organisms in cleanroom environments relevant to planetary protection. The revised section is in lines 143-146. This was briefly mentioned in the introduction (lines 87-90).

Line 137 –140: I do not understand what is happening here. You have total protein concentration and then how do you know that 50% of this is “pure” Rpf just by looking at gels? Are you looking at the size of bands? Is the size alone sufficient? I think you need to do a Western blot to prove that you are seeing Rpf. At the absolute minimum, you need to show the gels. If they are smears, you cannot draw conclusions. You also mention a final concentration of the Rpf, but I don’t know how you are isolating it or anything. This part needs to be re-written.

Line 142: Same as above. How are you getting the Rpf?

Response: the lysate was obtained from a BL21(DE3) *E. coli* strain overexpressing the *M. luteus* Rpf gene in a pNIC28 vector. After induction with IPTG and lysis by sonication, the lysate was clarified by centrifugation and filtered through a 0.2 μm filter. SDS-PAGE analysis showed a prominent band corresponding to the expected molecular weight of Rpf which we estimated to constitute ~50% of the total protein based on band intensity. We have now included the SDS-PAGE gel image as Supplementary Fig. S2 for reference.

While we acknowledge that Western blotting would provide a more definitive identification of Rpf, our current setup did not include a specific anti-Rpf antibody, which we note as a limitation. However, the gene construct and expression system are specific for *M. luteus* Rpf, and the observed bioactivity (resuscitation of dormant *T. phoenicis* and *M. luteus* cells) aligns well with expected functionality, supporting the identity of the expressed protein.

We also now clarify in both the figure legends and the Results section that the final concentration of the undiluted crude lysate used for the

resuscitation assays was $\sim 1 \mu\text{M}$ (27.5 $\mu\text{g/ml}$), calculated from the total protein content measured using a Nanodrop and the assumption of $\sim 50\%$ Rpf purity from gel densitometry. Serial dilutions of this lysate were then used in the growth recovery experiments, and the effects on lag phase and growth rate were quantified across conditions.

In our revised manuscript, we have revised the “Results” section to include a section “*Estimation of Rpf concentration in the crude lysate*” (155-167) and also expanded the “Materials and Methods” section (Lines 384-410) to provide additional details on the preparation and estimation of Rpf in the crude lysate.

Line 144: Was an entire column in a plate replicates? If so, where are the error bars in Figure 3? If not, why are you calling these columns? Same comments for lines 720 –726. Was your $n = 1$?

Response: This Figure (it is now Figure 2 in the revised version) shows the results of the initial screening for dormancy revival in LB and AMM. The plate has 4 Rows and 6 Columns of wells for each Row for a total of 24 wells. Thus, in Row A, Column 1 has cells with undiluted lysate (containing the Rpf), columns 2-5 had the cells with dilutions of the lysate ($10, 10^2, 10^3, 10^4$) and column 6 had the cells without any lysate (no Rpf).

We have included a Supplementary Figure S3 to show the layout of the plate for clarity. Each column in the 24-well plate represents a different lysate dilution condition, not replicates. For the initial dormancy revival screening shown in Figure 2, we assessed each condition in a single well (i.e., $n = 1$) due to constraints on lysate volume and the exploratory nature of this assay. Thus, the figure does not include error bars because it presents a representative result from a single screening experiment.

We have revised the figure legend and associated text (lines 170-172) to clarify that this was a preliminary screen conducted without technical or biological replicates.

Supplementary figure 2 does not convince me that the Rpf is responsible for anything. Also, you can graph the replicates (rows B –D) on the same graph with error bars instead of having separate graphs for each replicate. These are just biological replicates.

Response: After the initial screening without replicates, we repeated the experiment in triplicate for *T. phoenicis* in both LB and AMM media (Lines 205-218). In the revised version, the growth curves in LB are shown in Supplementary Figure S6. The purpose of Supplementary Figure S6 is to show how the strain responds to different liquid media (LB in this case) and to serve as a control for the minimal medium condition shown in Figure 3. While Supplementary Figure S6 alone may not conclusively demonstrate Rpf activity, it is intended to complement the data in Figure 3, where a crude lysate containing overexpressed Rpf (as shown in the accompanying gel) stimulates growth. Taken together, these figures support our interpretation that Rpf-containing lysate contributes to recovery from dormancy under nutrient-limited conditions.

These data were acquired using a Tecan SpectraFluor Plus instrument, with OD600 measurements taken every 15 minutes over an extended period, resulting in a large and dense dataset. The resulting data were processed via a custom Python script to generate the individual growth curves per replicate.

We fully understand the value of combining replicates into a single graph with error bars, and agree that this can enhance visual clarity. We have

now reprocessed and combined the data from the replicates (Supplementary Figure S6).

Figure 3: Change the X-axis from minutes. It is hard to get an understanding of what this means. Do it in hours. You should also mention how frequently a timepoint was measured.

Response: It has now been changed to hours. The frequency of timepoint measurement is now mentioned in Figure legends and in the revised text (lines 414-416).

You have a lot of data. Also, what is the “n” for this experiment? Did you repeat this? You should repeat to confirm if this was just an n of 1. If you did this in Supplemental figure 3, why is that supplemental and not the actual figure as replicate data is more reliable than an n of 1.

Response: Figure 3 (it is now Figure 2 in the revised version) represents an initial screen performed once (n=1) to identify key candidates. This screen was subsequently validated with independent biological replicates shown in Figure 3 (previously Supplemental Figure S3). We have moved these results into the main figure (Figure 3 in the revised version).

I also question the “spikiness” comment. Is this verified via microscopy? The issue is that the magenta and black lines in the third graph are suddenly “spiky” but they never reach log phase prior. Are the plates shaking at all to avoid this type of behavior (looks like yeast in methods/Line 343)? I think you should repeat this and do microscopy to prove that the aggregation is occurring (like in figure 5). You should also label these four graphs A –D so it is easier to follow.

Response: The observed “spikiness” in the magenta and black growth curves (now Figure 2 in the revised version) is consistent with our hypothesis of cellular aggregation. We included picture of the plate after the growth cycle as Figure 4. This shows visible clumps of cells supporting the notion of aggregation-induced OD600 fluctuation. Growth was conducted and measured (OD₆₀₀) every 15 minutes in a 24 well (4x6) microtiter plate with 1 ml of growth media in 2 ml wells, using a Tecan SpectraFluor Plus instrument. Continuous shaking was maintained between successive readings.

Supplementary Figure 3: do not even mention the dried wells in the legend as this is very confusing since there are no graphs. You should REPEAT this with more medium in the wells instead of reporting that they were dry. You cannot claim this as replicate data if there are no data.

Response: We agree that the reference to dried wells in the figure legend was unclear and have now removed this mention to avoid confusion. As clarified in the methods section, the growth was monitored using a Tecan SpectraFluor Plus instrument, which requires shaking in 1 mL volumes per well for optimal performance. Increasing the volume beyond this threshold compromises the instrument’s shaking efficiency and data quality. As a result, it was not technically feasible to repeat these conditions with a larger volume of medium. We have revised the figure (it is now Figure 3) and legend.

Line 151 –152: I do not understand this comment about low concentrations of Rpf. If you are referring to the overall achievement of higher ODs at later timepoints, I would say that more clearly and in which graphs, because it is not all of them.

Response: The differences in growth in terms of the OD₆₀₀ values observed (and the correct figures have been cited (Fig. 2 and Supplementary Figure S4) have now been mentioned in the text.

Figure 4: I don't think this adds much to the paper and should just be a supplement to Figure 3.

Response: This has now been included as a Supplementary Figure S5A-B

Supplementary figure 4: All the graphs look nearly identical. If B –D are triplicate, then graph them all together with error bars. This figure is also not mentioned in the Results section at all.

Response: The S4 figure (it is now Supplementary Figure S6) is described in the Results now (Lines 211-212).

Supplementary Figure 5: you mention Rpf from “crude” lysate. Is this protein not pure? How do we know that the data is based on the Rpf? Furthermore, I see no different between the Rpf dilutions expect for a very strange result with the black line. If Rpf was responsible, you would expect to see changes in data based on the concentration. But there is no trend in B that holds with the dilutions.

Response: As explained earlier, we overexpressed the *rpf* gene from *Micrococcus luteus* in *E. coli* and used the resulting crude lysate directly,

rather than the purified Rpf protein. This approach was consistent with previous reports of the use of crude lysate for reviving dormant cells supported by studies such as Ding *et al.* (2012) (Reference no. 82), which showed effective resuscitation of VBNC *Rhodococcus* sp. DS471. Another study explored the use of *M. luteus* supernatant containing Rpf to increase the cultivability of soil bacteria on oligotrophic media (Lopez Marin MA *et al.*, 2021- Reference no. 83).

Going forward, we are in the process of purifying the protein. We agree that using purified protein could allow for more precise dose-dependent interpretation, and we have now acknowledged this limitation in the revised discussion – lines 342-344.

Regarding Supplementary Figure 5B: we have corrected the figure (it is now Supplementary Figure S8) to clarify that the black line represents the no-Rpf (negative control) condition. As anticipated, this condition shows no growth stimulation. A modest increase in growth was seen relative to the control, consistent with previous reports that even lesser amounts of Rpf can have a biologically significant effect. We note that variability in activity may be due to the use of crude lysate, which can include background proteins and inconsistent Rpf concentration. These limitations are now stated in the revised version (lines 343-344).

We are currently optimizing purification protocols for future studies to characterize the dose response more precisely.

Line 205 –207: The comment about *Listeria* and *Vibrio* should be moved elsewhere.

Response: it is now moved to the Introduction, as suggested (Lines 94-96).

Line 210-215: There are possibly other explanations for what you observed. You need additional experimentation to confirm dormancy.

Response: We agree that dormancy is a complex physiological state with multiple contributing factors. In our study, we have taken a conservative approach in interpreting our data. Specifically, our observations that the population of cells remains viable (CFUs persist) despite a marked reduction in cultivability are consistent with a dormant-like state. Additionally, resuscitation in the presence of crude lysates containing overexpressed Rpf, is supportive of this interpretation.

While we acknowledge that additional experiments such as single-cell transcriptomics or metabolic flux analysis would provide further resolution, the current data—particularly the maintenance of viability in the absence of active growth and gene expression—are in line with operational definitions of dormancy in microbial systems. Lines 255-261.

Line 226 –227: You didn't do any expression experiments to support this claim.

Response: After desiccation, *T. phoenicis* cells lose cultivability and can be resuscitated in minimal media with the addition of Rpf containing crude lysate. The revival is after a prolonged time (approximately 50 hrs) without Rpf. These findings indicate that the cells are dormant, not dead – Lines 281-283.

Line 238 –239: What if the cells are just dying because you are drying them longer? There are other explanations than that the Rpf was inactive. You

would need to isolate the Rpf somehow to claim that it was degraded across a timeline.

Response: We agree that multiple factors can reduce cultivability including cell injury and death. While this is possible, our data support dormancy rather than cell death. *T. phoenicis* cells desiccated for 168 hours were still able to grow when Rpf was added, though with a longer lag phase—indicating dormancy, not death. This delay suggests deeper dormancy over time rather than Rpf degradation. While we used crude (not purified) Rpf, its ability to revive cell growth confirmed it remained functional. Future work will include purified Rpf to make better assessments – Lines 291-296.

Line 247 –255: Earlier you indicated that *M. luteus* is a commonly employed stand-in for *M. tuberculosis* studies. Seems to contradict with this claim. Likely they are just different organisms with different physiological responses. It seems like *M. luteus* is actually better at resisting damage due to desiccation. Also how repeatable is this? How many times did you do this to confirm a difference?

Response: Our intent in referencing *M. luteus* as a model organism for *M. tuberculosis* studies (Lines 108-111) was to highlight their shared ability to enter dormancy under nutrient starvation conditions, and the presence of key dormancy-associated genes such as those encoding resuscitation-promoting factors (Rpf). However, this does not imply that *M. luteus* and *T. phoenicis* would necessarily exhibit identical physiological responses under all stress conditions, such as desiccation.

In our desiccation experiments (Table 1), *M. luteus* did indeed show better initial resistance to viability loss compared to *T. phoenicis*, experiencing only a 10-fold reduction in cultivability after 48 hours versus a 10⁶-fold drop

observed for *T. phoenicis*. However, after 7 days (168 hours), both strains showed comparable loss in cultivability ($\sim 10^6$ -fold), suggesting that while *M. luteus* may resist short-term desiccation more effectively, it too becomes non-cultivable over longer durations. We interpret the rapid cultivability loss in *T. phoenicis* as an early dormancy response, not as cell death, based on its subsequent regrowth in AMM upon Rpf supplementation (Supplementary Fig. S8), consistent with a VBNC phenotype.

We have clarified in the manuscript (lines 304-306) that the SAF isolate *T. phoenicis*, having evolved in nutrient-limited and dry cleanroom conditions, may enter dormancy more rapidly under stress, while the *M. luteus* type strain used here (not a cleanroom isolate) lacks such environmental pre-adaptation. This is supported by studies of cleanroom isolates such as those cited in Cassilly et al. (2024) and Schultz et al. (2025), where some actinobacterial species (e.g., *Mycetocola manganoxydans*) exhibit increased desiccation and radiation resistance.

Furthermore, broader sampling and testing of actinobacterial strains from cleanrooms is essential to substantiate this observation. We now explicitly state this in the revised text (lines 315–322).

Line 278 –279: The methods are to detect spores which are not “active microbial components”

Response: changed as suggested – lines 338-339.

Line 322: Did you isolate the Rpf from *T. phoenicis*? If not, you MUST be clearer in the text that you were using *M. luteus* Rpf the entire time for both strains. Also, if you didn’t purify this protein, then you must call it a lysate or

crude. You should include more data in the supplemental to show that the protein has been isolated.

Response: We did not isolate the Rpf from *T. phoenicis*. In the revised version, we have made it clear we used *M. luteus* Rpf the entire time by overexpressing it in *E. coli* and used the lysate. The supplemental data shows the gel with the overexpressed Rpf. We have revised the manuscript to clarify that the resuscitation-promoting factor (Rpf) used in our experiments was derived from *Micrococcus luteus* and expressed in *Escherichia coli*, not isolated from *T. phoenicis*. We explicitly refer to the preparation as a crude intracellular lysate throughout the revised text to accurately reflect the use of unpurified Rpf-containing lysate.

As explained earlier, the use of crude lysate for reviving dormant cells is supported by studies such as Ding *et al.* (2012) (74), which showed effective resuscitation of VBNC *Rhodococcus* sp. DS471. Another study explored the use of *M. luteus* supernatant containing Rpf to increase the cultivability of soil bacteria on oligotrophic media (75).

Response to Reviewer #2 Comment:

“My primary remaining concern is that the results section remains largely qualitative, despite the data being collected quantitatively.. Please incorporate numerical values to enhance readability and interpretation.”

Response: We thank the reviewer for this valuable suggestion. In response, we have revised the Results section to include key quantitative values directly in the text where qualitative terms were previously used.

The revised text now provides the exact fold change and percentage reduction, making the result more precise and accessible without the need to consult Table 1. – Lines 148-150, 280-295.

Lines 141–142: "In each row, the well (column 1) that contained the highest concentration of Rpf grew faster, i.e., had the shortest growth lag phase."

State how much shorter the lag phase was.

Response: We have added specific lag phase durations to quantify the difference in growth – Lines 173-177.

Line 143 "Cell growth, in the absence of Rpf, (column 6) showed the greatest lag phase." Please provide a number for the lag phase so readers can compare it; do not make them have to check tables.

Response: We have added specific lag phase durations to quantify the difference in growth – Lines 178-181.

Line 143-~150 There are a lot of comparisons and referring to columns in the table. This makes it hard for a reader to absorb the data and critically think about it. Please add a few numbers and then use percents or fold change to allow readers to compared the data without having to stop and check the table each time. Words like "low" and "high" are relative and need data (numbers) in the text to support them.

Response: We have revised this entire section to data as suggested- Lines 168-191.

Lines 150-151 "Dormant cells of *T. phoenicis* in particular showed negligible growth in AMM in the absence of Rpf."

How was this determined? Statistics; if so please provide the test and p value. I understand you added more text here, but was this visual? We need to know assessment.

Response: Figure 2 shows the initial screening/evaluation which was not done in replicates (n=1). The growth was recorded based on the change in OD600 .

Following this screening, the effect of the Rpf on *T. phoenicis* was done in triplicates in LB and AMM. However, when grown in AMM, growth in two of the rows could not be observed as the wells containing the dilutions of the Rpf, dried up after 5000 minutes (Rows A and B). Therefore the values from the two remaining rows/wells (Supplementary Fig. S3) were taken and statistical values calculated (Lines 192-197).

This revision provides the statistical test used, and includes the p-value to confirm the significance of the result. The assessment was based on quantitative OD600 measurements, not visual inspection.

Re: Spectrum01692-25R1 (*Tersiccoccus phoenicis* (Actinobacteria) a spacecraft clean room isolate, exhibits dormancy)

Dear Prof. William Russell Widger:

In the revised file you answered the comments of the reviewers with respective changes.

Your manuscript has been accepted, and I am forwarding it to the ASM production staff for publication. Your paper will first be checked to make sure all elements meet the technical requirements. ASM staff will contact you if anything needs to be revised before copyediting and production can begin. Otherwise, you will be notified when your proofs are ready to be viewed.

Sincerely,
Dirk Tischler
Editor
Microbiology Spectrum